# Biogeographic Origin of *Kurixalus* (Anura, Rhacophoridae) on the East Asian Islands and Tempo of Diversification within *Kurixalus*

**DOI:** 10.3390/ani13172754

**Published:** 2023-08-30

**Authors:** Qiumei Mo, Tao Sun, Hui Chen, Guohua Yu, Lina Du

**Affiliations:** 1Key Laboratory of Ecology of Rare and Endangered Species and Environmental Protection (Guangxi Normal University), Ministry of Education, Guilin 541004, China; mo13978455206@163.com (Q.M.); suntao9658@126.com (T.S.); chen19126046382@163.com (H.C.); 2Guangxi Key Laboratory of Rare and Endangered Animal Ecology, College of Life Science, Guangxi Normal University, Guilin 541004, China

**Keywords:** jump dispersal, mainland origin, *Kurixalus*, East Asian islands

## Abstract

**Simple Summary:**

At present, there are two hypotheses about the biogeographic origin of *Kurixalus* on the East Asian islands. We reconstructed the ancestral distribution of *Kurixalus*, based on complete sampling and accurate selection of biogeographical analysis models. The results showed that *Kurixalus* on the East Asian islands have originated from the Asian mainland through two long-distance colonization events (jump dispersal). In addition, the analyses of the tempo of diversification revealed that the diversification rate of *Kurixalus* showed a slight decreasing trend. The relevant results will help us to comprehensively and accurately understand the geographical origin of *Kurixalus* and improve our understanding of the origin history of the flora and fauna of Taiwan Island.

**Abstract:**

The ancestral area of *Kurixalus* on the East Asian islands is under dispute, and two hypotheses exist, namely that distribution occurred only on the Asian mainland (scenario of dispersal) and that wide distribution occurred on both the Asian mainland and the East Asian islands (scenario of vicariance). In this study, we conducted biogeographic analyses and estimated the lineage divergence times based on the most complete sampling of species, to achieve a more comprehensive understanding on the origin of *Kurixalus* on the East Asian islands. Our results revealed that the process of jump dispersal (founder-event speciation) is the crucial process, resulting in the distribution of *Kurixalus* on the East Asian islands, and supported the model of the Asian mainland origin: that *Kurixalus* on the East Asian islands originated from the Asian mainland through two long-distance colonization events (jump dispersal), via the model of vicariance of a widespread ancestor on both the Asian mainland and the East Asian islands. Our results indicated that choices of historical biogeography models can have large impacts on biogeographic inference, and the procedure of model selection is very important in biogeographic analysis. The diversification rate of *Kurixaus* has slightly decreased over time, although the constant-rate model cannot be rejected.

## 1. Introduction

*Kurixalus* Ye, Fei, and Dubois, a genus of the family Rhacophoridae, inhabits various types of montane forests and has a wide distribution, from eastern India, eastward to Indochina, southern mainland China, and adjacent continental islands (Hainan, Taiwan, and Ryukyu), and southwards to Sundaland and the Philippine archipelago [1]. At present, a total of 23 species are recognized in the genus *Kurixalus* [1], namely: *K. silvaenaias* Hou, Peng, Miao, Liu, Li, and Orlov, 2021 [2], *K. motokawai* Nguyen, Matsui, and Eto, 2014 [3], *K. absconditus* Mediyansyah, Hamidy, Munir, and Matsui, 2019 [4], *K. lenquanensis* Yu, Wang, Hou, Rao, and Yang, 2017 [5], *K. chaseni* (Smith, 1924) [6], *K. banaensis* (Bourret, 1939) [7], *K. berylliniris* Wu, Huang, Tsai, Li, Jhang, and Wu, 2016 [8], *K. wangi* Wu, Huang, Tsai, Li, Jhang, and Wu, 2016 [8], *K. verrucosus* (Boulenger, 1893) [9], *K. raoi* Zeng, Wang, Yu, and Du, 2021 [10], *K. hainanus* (Zhao, Wang, and Shi, 2005) [11], *K. eiffingeri* (Boettger, 1895) [12], *K. bisacculus* (Taylor, 1962) [13], *K. naso* (Annandale, 1912) [14], *K. viridescens* Nguyen, Matsui, and Duc, 2014 [15], *K. odontotarsus* (Ye and Fei, 1993) [16], *K. appendiculatus* (Günther, 1858) [17], *K. inexpectatus* Messenger, Othman, Chuang, Yang, and Borzée, 2022 [18], *K. yangi* Yu, Hui, Rao, and Yang, 2018 [19], *K. idiootocus* (Kuramoto and Wang, 1987) [20], *K. baliogaster* (Inger, Orlov, and Darevsky, 1999) [21], *K. gracilloides* Nguyen, Duong, Luu, and Poyarkov, 2020 [22], and *K. pollicaris* (Werner, 1914 “1913”) [23]. Owing to the morphological conservation, the taxonomy of some *Kurixalus* species was once very conflicting, and disagreements over the taxonomic arrangement of some species existed among scholars. For instances, *K. hainanus* was once treated as *K. odontotarsus* by some authors (e.g., [24,25]) or a synonym of *K. bisacculus* [26], but Yu et al. [27] suggested that it is valid based on broad sampling. *Kurixalus verrucosus* was once grouped with *K. appendiculatus* by Wolf [28], and was then removed from the synonymy of *K. appendiculatus* by Inger et al. [21]. *Kurixalus chaseni* was grouped with *K. appendiculatus* by Smith [29] and was removed from the synonymy of *K. appendiculatus* by Matsui et al. [30], while *K. pollicaris* was grouped with *K. eiffingeri* by Zhao and Adler [31] and was recently resurrected by Dufresnes and Litvinchuk [32]. *Kurixalus wangi* and *K. berylliniris* were confused with *K. eiffingeri* [24] and were named as two independent species by Wu et al. [8], and *K. absconditus* was confused with *K. appendiculatus* [4] (Table 1). Therefore, previous molecular phylogenetic studies involving *Kurixalus* mainly focused on taxonomy, with little attention paid to the origin and dispersal history of *Kurixalus* (e.g., [33,34]).

Continental islands around the southeast edge of Asia have played an important role in creating novel evolutionary lineages and harboring historically diverse organisms [35]. The Taiwan Island and Ryukyu Islands lie on the edge of East Asia and are a part of the island-arc system along the western edge of the Pacific Ocean [36]. Previously, only two *Kurixalus* species were recorded from Taiwan and Ryukyu, namely *K. idiootocus* and *K. eiffingeri*, with the former being endemic to Taiwan and the latter being known from both Taiwan and Ryukyu [24,37]. Based on the morphology of eggs, tadpoles, and adults, mating calls, molecular data, ecology, and other information, Wu et al. considered that *K. eiffingeri* is a species complex and contains two cryptic species, namely *K. beryliniris,* being distributed in eastern Taiwan, and *K. wangi,* being distributed in southern Taiwan [8]. Recently, Dufresnes et al. proposed that the lineage consisting of *K. eiffengeri* from central and western Taiwan is also an independent species and referred it to *K. pollicaris* (Wernerm 1914 “1913”) [23,32]. Thus, currently, there are five *Kurixalus* species known from Taiwan and Ryukyu. Of them, four (*K. idiootocus*, *K. wangi*, *K. beryliniris*, and *K. pollicaris*) are endemic to Taiwan, and one (*K. eiffingeri*) is endemic to Ryukyu [1].

The earlier phylogenetic studies once recovered *Kurixalus* species on the East Asian islands (Taiwan and Ryukyu) to be monophyletic (e.g., [8,38,39]), which led to an understanding that *Kurixalus* on the East Asian islands has a single origin, until Yu et al. described *K. lenquanensis* from Yunnan Province, China [5]. This mainland species was recovered as the sister to *K. idiootocus*, implying that it may have originated from the Taiwan Island [5]. However, results of two recent biogeographic studies on *Kurixalus* have led to disputes on the origin of *Kurixalus* on the East Asian islands [22,40]. Yu et al. considered that the ancestor of Taiwanese *Kurixalus* originated from the Asian mainland via two long-distance colonization events (here referred to as a model of Asian mainland distribution) [40], whereas Nguyen et al. considered that the ancestor of species from Taiwan likely inhabited both Taiwan and the Asian mainland (here referred to as a model of wide distribution across the Asian mainland and East Asian islands) [22]. Probably, two factors may have contributed to these two incompatible biogeographic inferences. Firstly, the new lineage revealed by Yu et al. [40] (*K*. sp6, now known as *K. raoi* [10]) was not included in Nguyen et al.’s work [22], while the species described by Nguyen et al. [22] (*K. gracilloides*) was not included in Yu et al.’s study [40]. Secondly, the choice of the S-DIVA model in Nguyen et al.’s study was arbitrary, owing to the absence of the procedure of model selection [22]. It has been suggested that biogeographic analysis relying on a single DIVA model without the procedure of model selection can be potentially dangerous because this model leaves out the process of founder-event speciation (jump dispersal) [41], where at cladogenesis, one daughter lineage jumps to a new range outside the range of the ancestor (e.g., A->A, B).

Additionally, more recently, Hou et al. described *K. silvaenaias* from Sichuan Province, China [2], and Messenger et al. described *K. inexpectatus* from Zhejiang Province, China [18]. These two species are also closely related to *K. idiootocus* but have not yet been included in biogeographic analyses. Therefore, an analysis based on more comprehensive sampling is necessary to investigate the biogeographic history of *Kurixalus* and to test for the two hypotheses on the origin of *Kurixalus* on the east Asian islands mentioned above. Rapid speciation and adaptive radiation triggered by local adaptation and random drift often occur once an island is colonized [42]. Therefore, if *Kurixalus* has dispersed into Taiwan and Ryukyu from the Asian mainland, it also could be expected that the tempo of lineage accumulation of *Kurixalus* might have undergone an increase.

Herein, based upon the selection of the best biogeographic model and more complete sampling, we reconstructed the phylogenetic relationships and ancestral biogeographic areas of *Kurixalus* and estimated the lineage divergence times to achieve a more comprehensive understanding on the origin of *Kurixalus* on the East Asian islands. In addition, we analyzed the tempo of diversification in the genus to investigate whether *Kurixalus* has undergone an increase of the diversification rate.

## 2. Materials and Methods

### 2.1. Sample Collection

Available sequences of 12S rRNA, 16S rRNA, COI, recombination-activating gene 1 (RAG-1), tyrosinase (TYR), and brain-derived neurotrophic factor (BDNF) genes were obtained from GenBank for all known *Kurixalus* species, with the exceptions of *K. pollicaris* and *K. verrucosus* (Figure 1; Table 2), which is probably only distributed in Myanmar [27] and has never been sequenced. Homologous sequences of the three nuclear genes (RAG-1, TYR, BDNF) were sequenced for *K. lengquanensis*, *K. raoi*, and *K. silvaenaias* in this study using the primers and experimental protocols of Yu et al. [27], and all these new sequences have been deposited in GenBank under the Accession Numbers OQ719606–OQ719614. *Buergeria buergeri* (Temminck and Schlegel, 1895) [43] was included in the data as an outgroup following Yu et al. [40].

### 2.2. Phylogenetic Analysis

Sequences were aligned using MUSCLE with default parameters in MEGA 7 [44]. The sequence alignments were defined by genes, and then the best partitioning scheme and substitution models were selected in PartitionFinder v.2.1.1 [45] using the “greedy” algorithm [46]. Bayesian inference was performed in MRBAYES v3.2.6 [47], with substitution models for each partition. Two runs were performed simultaneously with four Markov chains, starting from a random tree. The chains were run for 3,000,000 generations and sampled every 100 generations. The first 25% of the sampled tree was discarded as burn-in after the standard deviation of split frequencies of the two runs was less than 0.01. The remaining trees were then used to create a consensus tree and to estimate the Bayesian posterior probabilities (BPPs).

### 2.3. Biogeographic Inference

We divided the present distribution of *Kurixalus* into three different biogeographic regions, including mainland Asia (1), Taiwan and Ryukyu (2), and Sundaland and the Philippines (3), according to Nguyen et al. [22], and then assigned each species to its own region. Biogeographic inference was conducted using the Bayesian stochastic search variable selection (BSSVS) [48] of the discrete phylogeographic model in BEAST version 1.8.0 [49], with the specification of the symmetric discrete trait substitution model. For this analysis, three different phylogenetic hypotheses were taken into account owing to the uncertainty of the phylogenetic position of *K*. *gracilloides* (see below). Combined with the inference of the ancestral area in BEAST, trees were calibrated in BEAST following Yu et al. [40]. Six independent runs were conducted for 7 × 10^8^ generations by sampling every 1000 generations for each phylogenetic hypothesis. The effective sample size values of parameters were confirmed in Tracer version 1.7.2 [50], and then trees produced by the runs were combined in LogCombiner version 1.8.0 [49]. The maximum clade credibility tree was constructed in TreeAnnotator version 1.8.0 [49].

We also estimated the ancestral ranges using the R package BioGeoBEARS [51] in RASP v.4.0 [52]. This package contains three widely used biogeographic models, namely dispersal-extinction-cladogenesis (DEC) [53], the likelihood version of dispersal-vicariance (DIVALIKE) [54], and the likelihood version of the BayArea model (BAYAREALIKE) [55], as well as three variants of them that concerned the jump dispersal by adding the parameter *j* (DEC+*j*, DIVALIKE+*j*, and BAYAREALIKE+*j*). We compared the fit of each model for each phylogenetic hypothesis using the AIC-weighted approach [56], and then ancestral ranges were estimated under the best-fit model. For this analysis, the ultra-metric time-calibrated trees generated from BEAST analyses were used. The maximum number of species unit areas was set to 3, and the number of random trees was set to 3000.

### 2.4. Diversification Analysis

To visualize the tempo of lineage accumulation during the history of *Kurixalus*, a lineage-through-time (LTT) plot was obtained using the analyses of phylogenetics and evolution (APE) [57].

We further tested for a significant departure from the null hypothesis of a constant rate of diversification using the constant-rate (CR) test [58], as implemented in the package APE. The statistic γ indicates whether internal nodes are closer to the root or to the tips of the tree than expected under a CR model (γ = 0). A significant *p*-value for a negative value of γ indicates a decrease in the diversification rate over time, and a positive value of γ indicates that nodes are closer to the tips and implies an acceleration of the accumulation of lineages. A CR model of diversification can be rejected at the 95% level of significance if γ < −1.645 [58]. Since we obtained a nearly complete sampling of species, the CR test was appropriate, without having to perform a Monte Carlo simulation to account for missing lineages.

We also analyzed the distribution of relative divergence times among species using the analysis of diversification with survival models [59], as implemented in the package APE. Three alternative models were tested: model A specifies a constant rate of diversification and represents the null hypothesis of no heterogeneity in rates over time, model B specifies that the diversification rate has gradually increased or decreased over time by estimating an additional parameter, β, and model C specifies two different diversification rates before (δ1) and after (δ2) a defined breakpoint at some time point in the past. For model B, values of β are less than one when the diversification rate increases over time, and greater than one when the rate decreases over time [59,60]. The diversification rate (birth rate minus death rate) was estimated using the method of Nee et al. [61] with the birth–death function in the package APE.

## 3. Results

### 3.1. Phylogenetic Relationship

Our gene fragments consisted of 388 bp from 12S rRNA, 835 bp from 16S rRNA, 770 bp from COI, 895 bp from RAG-1, 521 bp from TYR, and 608 bp from BDNF. The six genes were defined into five partitions, with the two ribosomal genes being defined as one (Table 3). Phylogenetic analysis based on the best partition scheme and substitution models of the combined data of the six genes strongly supported that *K. lenquanensis*, *K. idiootocus*, *K. silvaenaias*, *K. inexpectatus*, and *K*. *raoi* formed a clade (labeled as I), and that the other three species on the East Asian islands (*K. eiffingeri*, *K. berylliniris*, and *K. wangi*) also formed a clade (labeled as II). *Kurixalus gracilloides*, clade I, and clade II were grouped together with strong support, but the relationships between them were not resolved (Figure 2).

### 3.2. Ancestral Area Construction

Considering that the phylogenetic position of *K. gracilloides* was not yet resolved (Figure 2), we constructed three different phylogenetic hypotheses in BEAST to infer the ancestral area of *Kurixalus* on the East Asian islands: one assuming that *K. gracilloides* is the sister to clade I (H1; Figure 3b,e), one assuming that *K. gracilloides* is the sister to clade II (H2; Figure 3c,f), and one assuming that *K. gracilloides* is the sister to clades I and II (H3; Figure 3d,g). In all cases, the ancestral area of *Kurixalus* on the East Asian islands was estimated to be the Asian mainland, and two colonization events (jump dispersal) from mainland Asia to the East Asian islands were identified by the BSSVS analyses, one for the ancestor of *K. eiffingeri*, *K. berylliniris*, and *K. wangi,* and one for the lineage giving rise to *K. idiootocus* (Figure 3b–d). A similar biogeographic origin and process were also inferred by the BioGeoBEARS analyses based on the model of DIVALIKE+*j* (Figure 3e–g), which was selected as the best biogeographic model for the genus *Kurixalus* (Table 4).

### 3.3. Tempo of Diversification

The LTT plot was almost straight until ca. 5 Mya, and then the slope slightly decreased (Figure 4), indicating that the rate of lineage accumulation slightly decreased since ca. 5 Mya. The survivorship analysis favored model B (AIC = 91.32, Table 5), and the value of β was greater than 1 (β = 1.53), also indicating that the rate of diversification decreased over time. The CR test computed a negative gamma value (γ = −1.01), indicating a decrease in net diversification rates over time, although it was not significant (*p* = 0.31).

## 4. Discussion

Taiwan mostly acquired its fauna from the Eurasian mainland since it emerged during the Late Miocene or early Pliocene [62,63]. In this study, our results of BSSVS and BioGeoBEARS analyses based on the best-fit biogeography model (DIVALIKE+*j*) also strongly supported that *Kurixalus* immigrated to Taiwan from the Asian mainland through two long-distance colonization events (jump dispersal). The common ancestor of *K. eiffingeri*, *K. berylliniris*, and *K. wangi* (clade II) dispersed to Taiwan at ca. 5 Mya and then diverged into different species from ca. 3.2 Mya, and the lineage giving rise to *K. idiootocus* also dispersed to Taiwan from ca. 1 Mya (Figure 3). This result is in congruence with the inference of Yu et al. [40] and matches with the geological evidence that proto-Taiwan Island emerged from water in the late Miocene (ca. 6.5 Mya) [64] or early Pliocene (4–5 Mya) [65]. The land–bridge connection across the Taiwan Strait occurred after 2.6 Mya [40,66,67]. Therefore, as assumed by Yu et al. [40], initially, the genus *Kurixalus* likely colonized Taiwan by way of transoceanic dispersal, which might be the major pathway to disperse to Taiwan for vertebrate animals [63]. More studies are necessary to obtain the precise timing of the two colonization events because the phylogenetic relationships between clade I, clade II, and *K. gracilloides* were not resolved here (Figure 2).

Different historical biogeography models involve different assumptions about the processes that have produced the geographic ranges [68], and it has been suggested that the process of founder-event speciation is crucial, especially in island systems [69,70,71,72]. When we used the pure DIVA model (DIVALIKE) in the biogeographic analysis, it was inferred that two vicariance events of ancestors, widely distributed on both the Asian mainland and East Asian islands, induced the speciation of *Kurixalus* on the East Asian islands (Figure 5). The difference in the biogeographic inferences that resulted from the DIVALIKE+*j* and pure DIVA (DIVALIKE) models indicates that the choice of the historical biogeography model can have a large impact on biogeographic inference, and that the process of jump dispersal (founder-event speciation) is also the crucial process resulting in the distribution of *Kurixalus* on the East Asian islands. The pure DIVA method is, on the whole, biased towards vicariance, and hence tends to result in the ancestors being spuriously inferred as widespread owing to the fact that the process of jump dispersal is not assumed by the DIVA model [18,72].

Additionally, the scenario of jump dispersal was favored over the scenario of vicariance for the clades of the East Asian islands in all three phylogenetic hypotheses (Figure 3), indicating that assumptions about the process can have a much larger impact on conclusions about the biogeographical history of oceanic clades than differences in phylogenetic topology. This result further highlights the importance of the procedure of model selection in biogeographic inference.

Contrary to the expectation that the tempo of lineage accumulation of *Kurixalus* might have undergone an increase, the analyses of diversification revealed that the diversification rate of *Kurixalus* has slightly decreased since ca. 5 Mya, although the model of constant diversification was not rejected by the CR test. Possibly, the filling of geographical and/or ecological space during late history [73], which decreases the likelihood of geographical speciation through range subdivision [74], might have contributed to this slight change in the diversification rate of the genus *Kurixalus*.

## 5. Conclusions

In summary, the results of this study supported the hypothesis that *Kurixalus* originated from the Asian mainland. Based on these data, this study emphasized that the process of jump dispersal is the key to the distribution of *Kurixalus* on the East Asian islands and stressed that the choice of the historical biogeography model had a significant impact on the biogeographic inference. Additionally, the genus *Kurixalus* might have undergone a slight decrease in the diversification rate over time, although the constant-rate model cannot be fully rejected.

## Figures and Tables

**Figure 1 animals-13-02754-f001:**
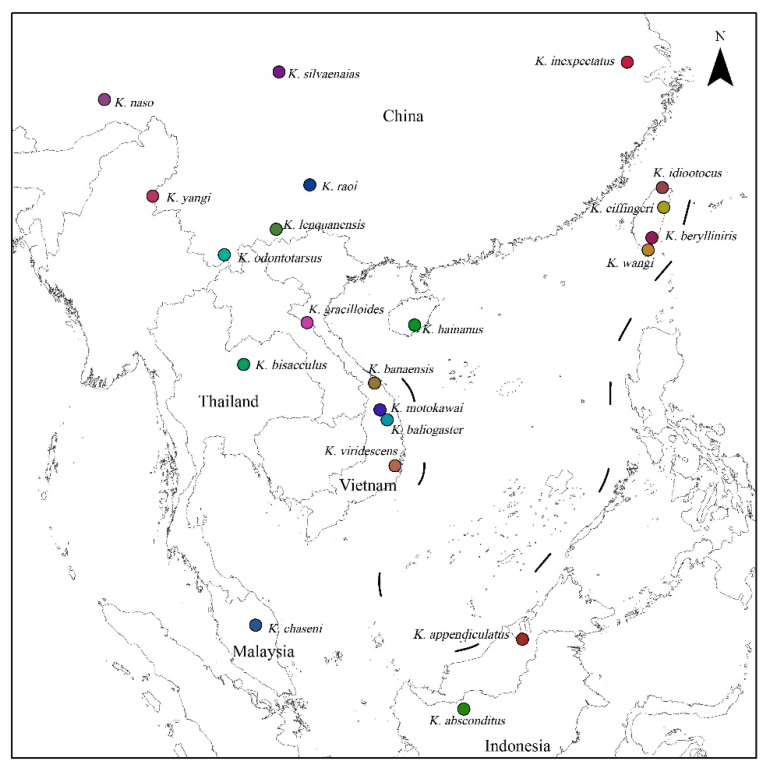
Sampling localities of this study.

**Figure 2 animals-13-02754-f002:**
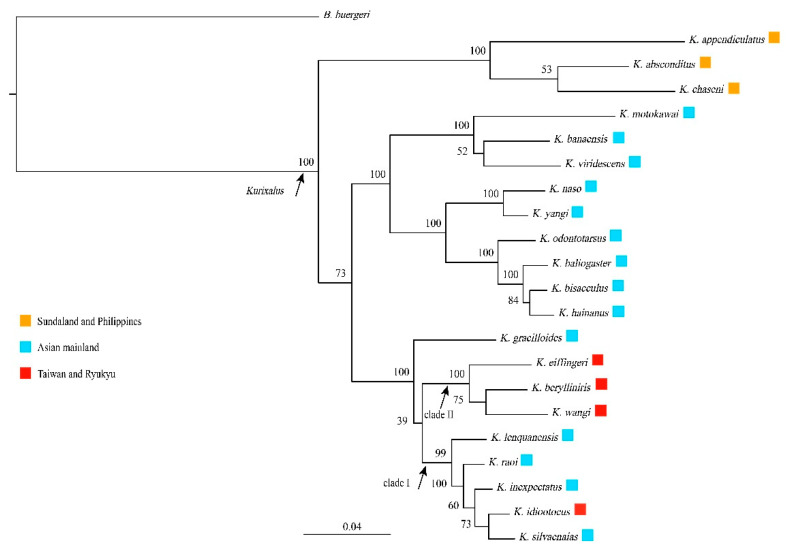
Bayesian phylogram of the genus *Kurixalus* based on the combined data of three mitochondrial genes (12S rRNA, 16S rRNA, COI) and three nuclear gene sequences (TYR, RAG-1, BDNF). Numbers above branches are the Bayesian posterior probabilities. The biogeographic assignments of each species are highlighted with different colors.

**Figure 3 animals-13-02754-f003:**
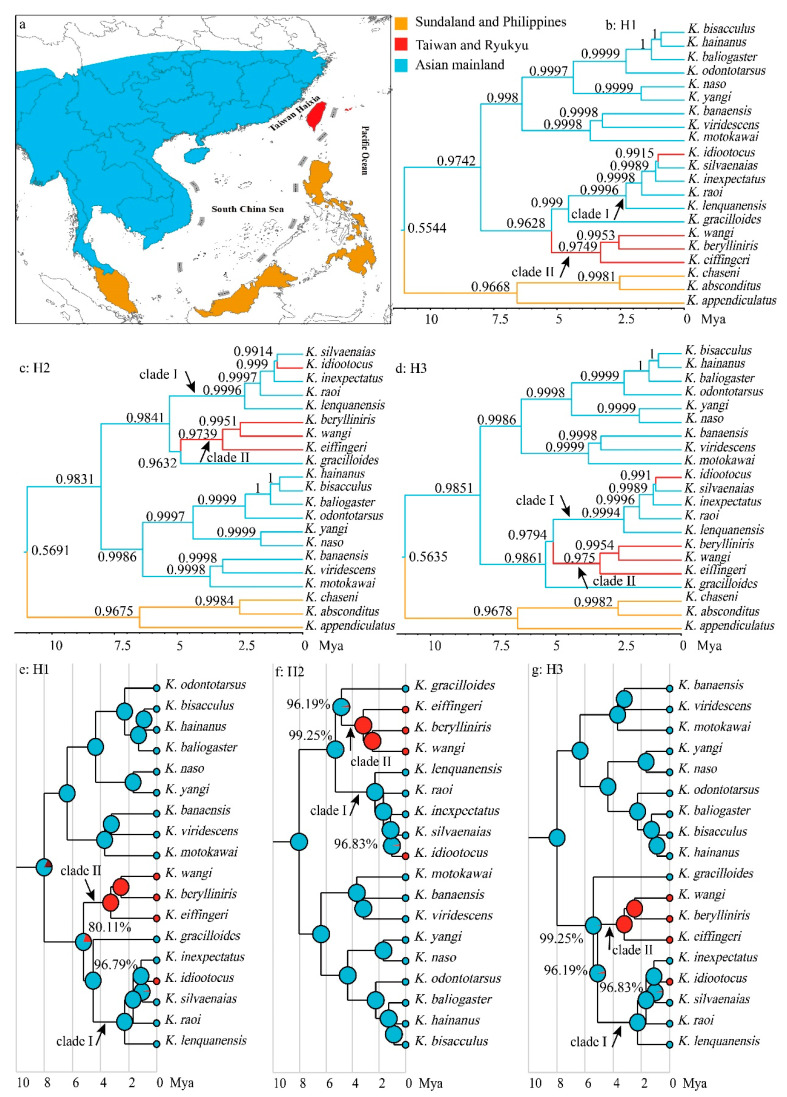
Biogeographic division (**a**) and ancestral area inferences resulted from BSSVS analyses (**b**–**d**) and the best-fit biogeography model (DIVALIKE+*j*); (**e**–**g**). Three different phylogenetic hypotheses (H1–H3) were taken into account because the relationships between *K. gracilloides*, clade I, and clade II were not resolved. Numbers above branches and near the nodes are the probabilities of the reconstructed ancestral area.

**Figure 4 animals-13-02754-f004:**
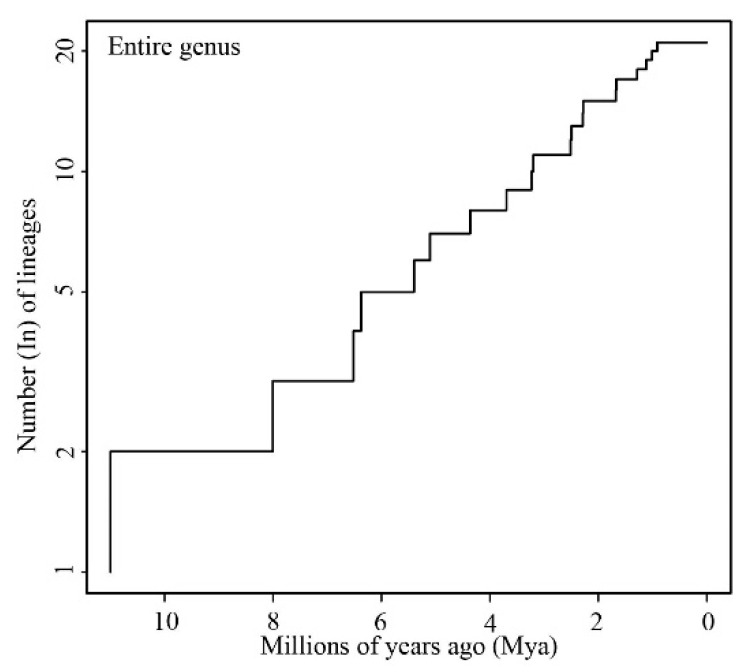
The lineage accumulation over time for the entire *Kurixalus* genus.

**Figure 5 animals-13-02754-f005:**
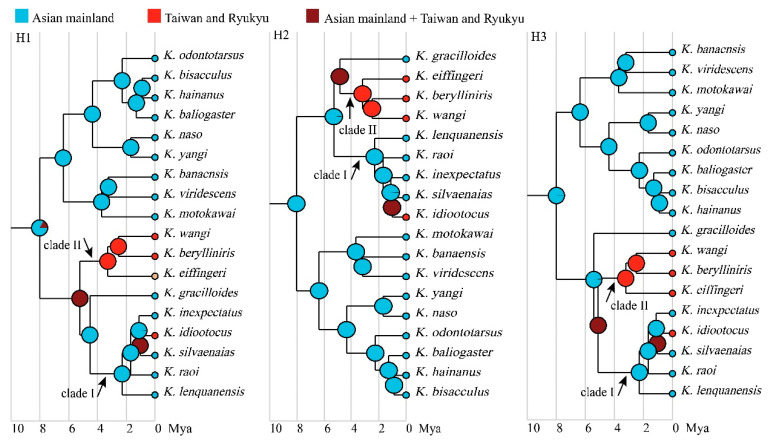
Biogeographic inferences resulting from pure DIVA model (DIVALIKE model) for the three different phylogenetic hypotheses.

**Table 1 animals-13-02754-t001:** Disputes on the taxonomy of *Kurixalus.*.

Species	Resources/Species			
	Mediyansyah et al. [4]	Mediyansyah et al. [4]		
*K. absconditus* [4]	*K. appendiculatus*	*K. absconditus*		
	Smith [29]	Matsui et al. [30]		
*K. chaseni* [6]	*K. appendiculatus*	*K. chaseni*		
	Fei [24]	Wu et al. [8]		
*K. berylliniris* [8]	*K. eiffingeri*	*K. berylliniris*		
	Fei [24]	Fei et al. [25]	Yu et al. [26]	Yu et al. [27]
*K. hainanus* [11]	*K. odontotarsu*	*K. odontotarsu*	*K. bisacculus*	*K. hainanus*
	Zhao et al. [31]	Dufresnes et al. [32]		
*K. pollicaris* [23]	*K. eiffingeri*	*K. pollicaris*		
	Fei [24]	Wu et al. [8]		
*K. wangi* [8]	*K. eiffingeri*	*K. wangi*		
	Wolf [28]	Inger et al. [21]		
*K. verrucosus* [9]	*K. appendiculatus*	*K. verrucosus*		

**Table 2 animals-13-02754-t002:** Species used in this study (*B*. = *Buergeria*, *K*. = *Kurixalus*).

Species	Voucher Number	Locality	12S	16S	COI	RAG-1	TYR	BDNF
*B. buergeri*	KUHE 13260	Japan	AB127977	AB127977	AB127977	AB728271	AB728322	AB72821
*K. absconditus*	MZB Amph 21862	Indonesia	-	MN727052	-	-	-	-
*K. appendiculatus*	KU 324192 NMBE 1056476 FMNH 267904	Bohol, Philippines Malaysia	KF933206	-	-	JQ060911	KC961232	KC961139
*K. chaseni*	FMNH 267896	Malaysia	JQ060948	JQ060937	KX554539	-	-	-
*K. baliogaster*	ROM 29860 ROM 29862	Vietnam	KX554475	KX554537	KX554647	KX554853	KX554740	KX554921
*K. banaensis*	ROM 32986	Vietnam	GQ285667	GQ285667	-	GQ285752	GQ285799	GQ285689
*K. berylliniris*	11311	Taiwan, China	-	DQ468669	DQ468677	-	-	-
*K. bisacculus*	THNHM 10051 KUHE 19333	Thailand	GU227279	GU227334	KX554633	KX554850	KX554737	KX554918
*K. eiffingeri*	11333 UMFS 5969	Taiwan, China	-	DQ468670	DQ468678	-	DQ28293	-
*K. gracilloides*	SIEZC 30189	Vietnam	-	MN510865	-	-	-	-
*K. hainanus*	YGH 090044 HNNU A1180	Yunnan, China Hainan, China	GU227248	GU227299	KX554599	GQ285749	EU215608	GQ285686
*K. idiootocus*	A127 SCUM 061107L	Taiwan, China	-	DQ468674	DQ468682	GQ285751	EU215607	GQ285688
*K. lenquanensis*	YGH 20160036	Yunnan, China	MK348042	KY768931	MK348050	OQ719606	OQ719609	OQ719612
*K. motokawai*	VNMN 03458	Vietnam	LC002888	LC002888	-	-	-	-
*K. naso*	Rao 06301	Tibet, China	KX554422	KX554484	KX554547	KX554745	KX554653	-
*K. odontotarsus*	YGH 090131 SCUM 060688L	Yunnan, China	GU227240	GU227290	KX554576	GQ285750	EU215609	GQ285687
*K. viridescens*	VNMN 03802	Vietnam	AB933284	AB933284	-	-	-	-
*K. wangi*	11328	Taiwan, China	-	DQ468671	DQ468679	-	-	-
*K. yangi*	Rao 14102901 Rao 14102908	Yunnan, China	KX554429	KX554491	KX554557	KX554761	KX554666	KX554863
*K. raoi*	YU1406033	Guizhou, China	MK348044	MK348047	MK348052	OQ719607	OQ719610	OQ719613
*K. silvaenaias*	CIB118049	Sichun, China	-	OL898656	OL854130	OQ719608	OQ719611	OQ719614
*K. inexpectatus*	NJFU20180704001 NJFU20180706003	Zhejiang, China	MW115094	-	-	-	MW148403	-

**Table 3 animals-13-02754-t003:** Partitioning strategy and the best substitution model for each partition.

Partitions	Best Model
(1) 12S rRNA and 16S rRNA	GTR+I+G
(2) BDNF	GTR+I
(3) COI	GTR+G
(4) RAG-1	GTR+I+G
(5) TYR	K80+G

**Table 4 animals-13-02754-t004:** Comparison of the six models of the ancestral area estimation of *Kurixalus*.

H	Model	Ln *L*	*n*	*d*	*e*	*j*	AICc	AICc_wt
H1	DEC	−14.73	2	0.0044	1.00 × 10^−12^	0	34.13	0.013
	DEC+*j*	−10.19	3	8.80 × 10^−10^	8.80 × 10^−10^	0.029	27.79	0.31
	DIVALIKE	−13.97	2	0.012	1.00 × 10^−12^	0	32.61	0.027
	**DIVALIKE+*j***	**−9.53**	**3**	**1.00 × 10^−12^**	**1.00 × 10^−12^**	**0.032**	**26.48**	**0.59**
	BAYAREALIKE	−23.6	2	0.014	0.055	0	51.86	1.80 × 10^−6^
	BAYAREALIKE+*j*	−11.79	3	1.00 × 10^−7^	1.00 × 10^−7^	0.044	30.99	0.062
H2	DEC	−16.74	2	0.0082	1.00 × 10^−12^	0	38.15	0.0022
	DEC+*j*	−10.52	3	1.00 × 10^−12^	1.00 × 10^−12^	0.033	28.45	0.28
	DIVALIKE	−15.98	2	0.012	1.00 × 10^−12^	0	36.62	0.0048
	**DIVALIKE+*j***	**−9.71**	**3**	**1.00 × 10^−12^**	**1.00 × 10^−12^**	**0.031**	**26.84**	**0.63**
	BAYAREALIKE	−23.88	2	0.014	0.055	0	52.42	1.80 × 10^−6^
	BAYAREALIKE+*j*	−11.83	3	1.00 × 10^−7^	1.00 × 10^−7^	0.044	31.06	0.077
H3	DEC	−15.03	2	1.00 × 10^−12^	1.00 × 10^−12^	0	34.72	0.012
	DEC+*j*	−10.52	3	1.30 × 10^−9^	1.60 × 10^−10^	0.033	28.45	0.28
	DIVALIKE	−16.43	2	0.012	2.00 × 10^−9^	0	37.53	0.003
	**DIVALIKE+*j***	**−9.71**	**3**	**1.00 × 10^−12^**	**2.40 × 10^−9^**	**0.031**	**26.84**	**0.63**
	BAYAREALIKE	−23.76	2	0.014	0.054	0	52.19	2.00 × 10^−6^
	BAYAREALIKE+*j*	−11.83	3	1.00 × 10^−7^	1.00 × 10^−7^	0.044	31.06	0.076

Note: Ln *L*, log-likelihood; *n*, number of parameters; *d*, rate of dispersal; *e*, rate of extinction; *j*, likelihood of founder-event speciation at cladogenesis; AICc, corrected Akaike information criterion. The AICc_wt was used to compare all the models to select the best one. The preferred model is indicated in bold.

**Table 5 animals-13-02754-t005:** Test of constant diversification of *Kurixalus* with survival models.

Ln *L* (AIC)			γ (*p*-Value)
Model A	Model B	Model C	
−46.204 (94.407)	−43.66 (91.32)	−45.632 (95.265)	−1.01 (0.31)

Note: Model A vs. Model B: χ^2^ = 5.09, df = 1, *p* = 0.02. Model A vs. Model C: χ^2^ = 1.14, df = 1, *p* = 0.29.

## Data Availability

Sequence data used in this study are accessible from GenBank (https://www.ncbi.nlm.nih.gov/genbank/).

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
