# Peer review of "Biogeographic Origin of Kurixalus (Anura, Rhacophoridae) on the East Asian Islands and Tempo of Diversification within Kurixalus"

_animals, 2023, doi:10.3390/ani13172754_

Round 1

Reviewer 1 Report

In the manuscript “Biogeographic origin of Kurixalus (Anura, Rhacophoridae) on the East Asian islands and tempo of diversification within Kurixalus”, Qiumei Mo et al. used existing sequences already available to analyse the origin of Kurixalus. The authors showed that the choices of historical biogeography models can have large impacts on biogeographic inference. Although the manuscript is clear, here are some comment to improve it :

Introduction: At the begining ot the introduction there is an issue in the citations that makes it very difficult to read. However, regardless this issue, this section looks like a list very difficult to follow for a reader. One or several tables would be very helpful in the introduction to recapitulate some important informations. Adding a map were the species are originated from (if reasonably possible when the information is available) could also help to visualize the species more easily.

Line 118: the name of the paragraph "data preparation" does not fit the content of the paragraph. The authors should reconsider it for another title.

Line 192: The authors should define the abbreviation APE.

Figures : The authors should consider increasing the quality of the figures.

Conclusion: The conclusion is almost identical to the abstract. The authors should consider to give more insight in the conclusion or at least to reformulate a bit more the sentences.

Some part can be improved a bit but the overall english is ok.

Reviewer 2 Report

The paper sets out to analyse the biogeographic origin of the Kurixalus species on East Asian islands, using previously described functions of the BEAST software; a phylogeny based on single specimens of all known species of this genus (except one). A jump dispersal explanation for the presence of Kurixalus on Taiwan and Ryukyu is proposed, and this is supported by the analyses presented here.

Potential weaknesses of the study are the failure of the phylogeny to resolve K. gracilloides, and the stated existence of cryptic species currently grouped under existing species. The first of these is dealt with adequately, in my view, by presenting alternative scenarios and showing that the biogeographic inference is the same in each of these.  It would be helpful if the authors could discuss how possible cryptic species could influence the conclusions given here, and what research would be needed to resolve this.

As far as my understanding of the methodology goes, it appears robust.

The lineage through time (LTT) data is visually not very impressive but the statistical results can be accepted. However, the authors do not discuss the relevance of LTT to the biogeographical conclusions, and I wonder whether this really merits its mention in the title of the paper, unless its connection to the biogeographic result is shown to be useful.

The paper is generally well written but there are a number of minor editorial points to be corrected later.
